# An integrated multiple driver mesocosm experiment reveals the effect of global change on planktonic food web structure

Hugo Duarte Moreno [1✉], Martin Köring[1], Julien Di Pane [1], Nelly Tremblay [1], Karen H. Wiltshire[1,2], Maarten Boersma [1,3] & Cédric L. Meunier [1]

Global change puts coastal marine systems under pressure, affecting community structure and functioning. Here, we conducted a mesocosm experiment with an integrated multiple driver design to assess the impact of future global change scenarios on plankton, a key component of marine food webs. The experimental treatments were based on the RCP 6.0 and 8.5 scenarios developed by the IPCC, which were Extended (ERCP) to integrate the future predicted changing nutrient inputs into coastal waters. We show that simultaneous influence of warming, acidification, and increased N:P ratios alter plankton dynamics, favours smaller phytoplankton species, benefits microzooplankton, and impairs mesozooplankton. We observed that future environmental conditions may lead to the rise of *Emiliania huxleyi* and demise of *Noctiluca scintillans*, key species for coastal planktonic food webs. In this study, we identified a tipping point between ERCP 6.0 and ERCP 8.5 scenarios, beyond which alterations of food web structure and dynamics are substantial.

[1] Alfred-Wegener-Institut, Helmholtz-Zentrum für Polar- und Meeresforschung, Biologische Anstalt Helgoland, Ostkaje 1118, 27498 Helgoland, Germany. [2] Alfred-Wegener-Institut, Helmholtz-Zentrum für Polar- und Meeresforschung, Wattenmeerstation, Hafenstr. 43, 25992 List auf Sylt, Germany. [3] University of Bremen, FB 2, Bibliothekstr. 1, 28359 Bremen, Germany. ✉email: hugo.moreno@awi.de

Human activities and associated increasing greenhouse gas emissions have caused simultaneous changes in a range of marine abiotic parameters. The Intergovernmental Panel on Climate Change (IPCC) established different scenarios projecting that, depending on humanity's effort to reduce greenhouse gas emissions, by 2100, the temperature may increase by 1–6 °C and pH may decrease by 0.1–0.4 units in the ocean's upper layers[1]. In addition, urban, agricultural and industrial development will continue to alter biogeochemical cycles through nutrient runoffs, increasing phosphorus limitations in European coastal marine systems[2]. Consequently, marine organisms are currently, and will continue to be, exposed to the simultaneous effects of multiple anthropogenic drivers. The pressure exerted by these changes on coastal marine systems threatens biological community structure and food web functioning[3,4]. Planktonic organisms are particularly sensitive to ecosystem change, and, given their central role in marine food webs, these organisms are of vital importance for ecosystem health[5]. Despite the urgent need to understand and predict how global change will influence planktonic food webs, there is still a striking paucity of information on the integrated impact of multiple drivers, especially in a community context. The few studies addressing the combined effects on plankton communities showed, for example, negative effects on copepod abundance, as well as shifts in phytoplankton organismal size[6–9].

Among the different methods that can be employed to address community responses to multiple global change drivers experimentally and mechanically, mesocosm approaches provide the highest level of ecological relevance while still being conducive to experimental manipulations[10]. By incorporating natural assemblages and by addressing mechanistic relationships across trophic levels that take place in complex natural systems, mesocosms go beyond small, tightly controlled experiments which suffer from limited realism[11]. The main limitation of the mesocosm approach is the difficulty of replication, due to the high costs of acquiring and maintaining such systems[10]. For this reason, full-factorial mesocosm experiments are scarce. Although understanding the individual effect of global change drivers, such as temperature, pH or dissolved nutrient concentrations, on the functioning of planktonic communities can inform specific mitigation strategies, it is important to consider that these drivers are simultaneously changing in natural environments. Hence, we applied an integrated multiple driver design to assess the potential impact of global change on natural coastal plankton communities. We tested the influence of two future scenarios against current environmental conditions in triplicates: the Ambient condition (ambient temperature and pH) and the Representative Concentration Pathway 6.0 (RCP 6.0, +1.5 °C, −0.2 pH) and RCP 8.5 (+3.0 °C, −0.3 pH), proposed by the IPCC for 2100[1]. In addition, as nutrient inputs are also predicted to change towards considerably higher nitrogen to phosphorus ratios (N:P) in coastal seas, especially those in Europe[2], we extended the RCP scenarios (ERCP) to simulate changing nutrient regimes, with a N:P ratio (molar) of 16 (Redfield ratio) for the Ambient scenario and 25 for both future scenarios (ERCP 6.0 and 8.5). It is currently of utmost importance to make accurate and reliable predictions of the fate of planktonic communities in future conditions. Although our experimental design does not enable to draw conclusions about individual drivers effect, we believe that our work provides a more realistic assessment of these drivers' impact than an experiment addressing drivers singly would.

The mesocosm experiment was conducted over three weeks in late summer (August–September) 2018. Seawater containing a natural plankton community was collected from the coastal North Sea. At the onset of the experiment, $CO_2$ saturated seawater was added to the ERCP scenario mesocosms to adjust $pCO_2$ and pH levels for each scenario. To create a realistic environment, we also manipulated the atmospheric $pCO_2$ in the enclosed mesocosm tanks throughout the experiment. Seawater temperature was adjusted daily according to the current North Sea temperature measured at the Helgoland Roads for the Ambient, and 1.5 °C and 3.0 °C warmer for the ERCP 6.0 and ERCP 8.5 scenarios, respectively. Dissolved nutrient concentrations were determined at the onset of the experiment and adjusted to reach the desired N:P ratios. Samples were taken in an interval of 1–3 days depending on the phytoplankton bloom development, and community composition, except for the large mesozooplankton, was monitored throughout the experiment period. Across scenarios, no significant difference was found in biomass of phytoplankton, microzooplankton and bacterioplankton on the first day of the experiment (Kruskal–Wallis test, df = 2, $P > 0.05$). The effects of the ERCP scenarios on plankton community biomass were statistically assessed through the likelihood ratio test (LRT), and principal response curve (PRC) analysis was applied to identify the influence of the ERCP scenarios on community composition.

## Results and discussion

**Overall**. In all treatments, we observed a first phytoplankton bloom, which lasted roughly 10 days, followed by a second bloom of different magnitude and composition between the treatments. We observed that, throughout the experiment, the planktonic food web was relatively similar in the Ambient treatment and in the ERCP 6.0 scenario, whereas the ERCP 8.5 scenario substantially altered the biomass, structure, and dynamics of multiple trophic levels (Fig. 1). The ERCP 8.5 scenario benefited the emergence of nanophytoplankton, specifically coccolithophores, at the expense of larger diatoms, especially in the second bloom. This has implications for the marine carbon pump due to the calcification capacity of coccolithophores[12]. Mesozooplankton biomass was largely reduced in the ERCP 8.5 scenario, whilst the biomass of microzooplankton was higher in this treatment than in the other two. The increase of micrograzers and lower mesozooplankton biomass are indicative of a microbial loop dominance in this future scenario, and of a potential diminution of energy transfer to higher trophic levels. We wish to note that, due to the relatively short duration of the experiment, this study does not consider the potential adaptation of planktonic communities that may take place over longer periods of time.

**The rise of nanophytoplankton**. Total cumulative phytoplankton biomass was not affected by the experimental treatment (GLM, df 86, ERCP 6.0 $P = 0.90$, ERCP 8.5 $P = 0.17$, $n = 3$, Fig. 2a). It appeared that the timing in phytoplankton biomass was also not statistically different among treatments (LRT; df 86, $P = 0.46$). Phytoplankton biomass increased exponentially from the beginning of the experiment in all treatments to reach a stationary phase on day 4. During this first phytoplankton bloom, we observed a gradient in the relative abundance of the large diatom *Guinardia flaccida* (GLM, df 86, ERCP 6.0 $P = 0.01$, ERCP 8.5 $P < 0.0001$, $n = 3$) from high in Ambient, to lower in ERCP 6.0 and ERCP 8.5, and the opposite in the contribution of nanophytoplankton (<20 μm) to the total phytoplankton biomass (GLM, df 86, ERCP 6.0 $P = 0.88$, ERCP 8.5 $P = 0.04$, $n = 3$, Figs. 1 and 2b). During this first bloom, both ERCP scenarios yielded lower phytoplankton biomass and were largely favourable towards nanophytoplankton at the expense of larger microalgal species (Supplementary Fig. 1). This result is similar to previous studies showing a negative effect of warming and acidification on the mean cell size of phytoplankton communities[13,14], which can be exacerbated when nutrient availability is low[15,16].

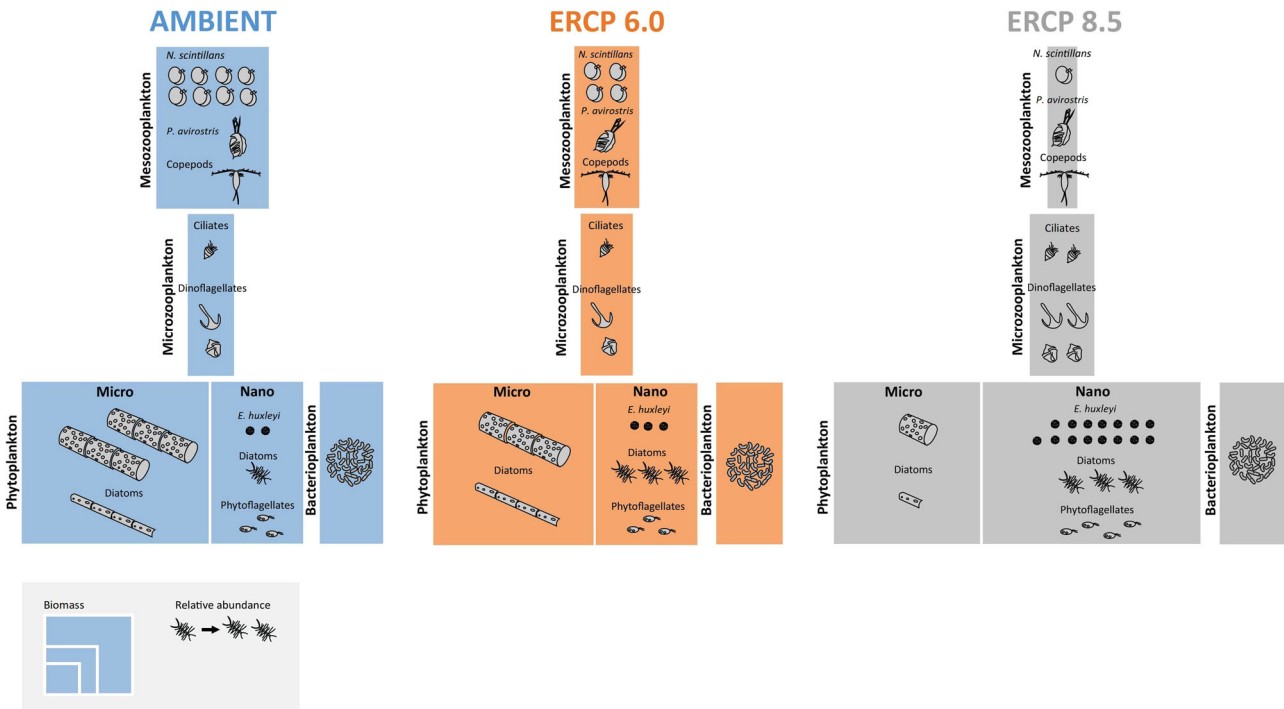

**Fig. 1 Infographic of biomass and dominant taxa for different food web compartments under the Ambient treatment and the ERCP scenarios.** Colours represent the Ambient treatment and Extended Representative Concentration Pathway (ERCP) scenarios (blue = Ambient, orange = ERCP 6.0, grey = ERCP 8.5), box size represents the total biomass of each compartment, and the number of individuals represents the relative abundance of taxonomic groups within a scenario. Phytoplankton biomass is divided between microphytoplankton (>20 µm) and nanophytoplankton (<20 µm). Plankton biomass and relative abundance are displayed to scale.

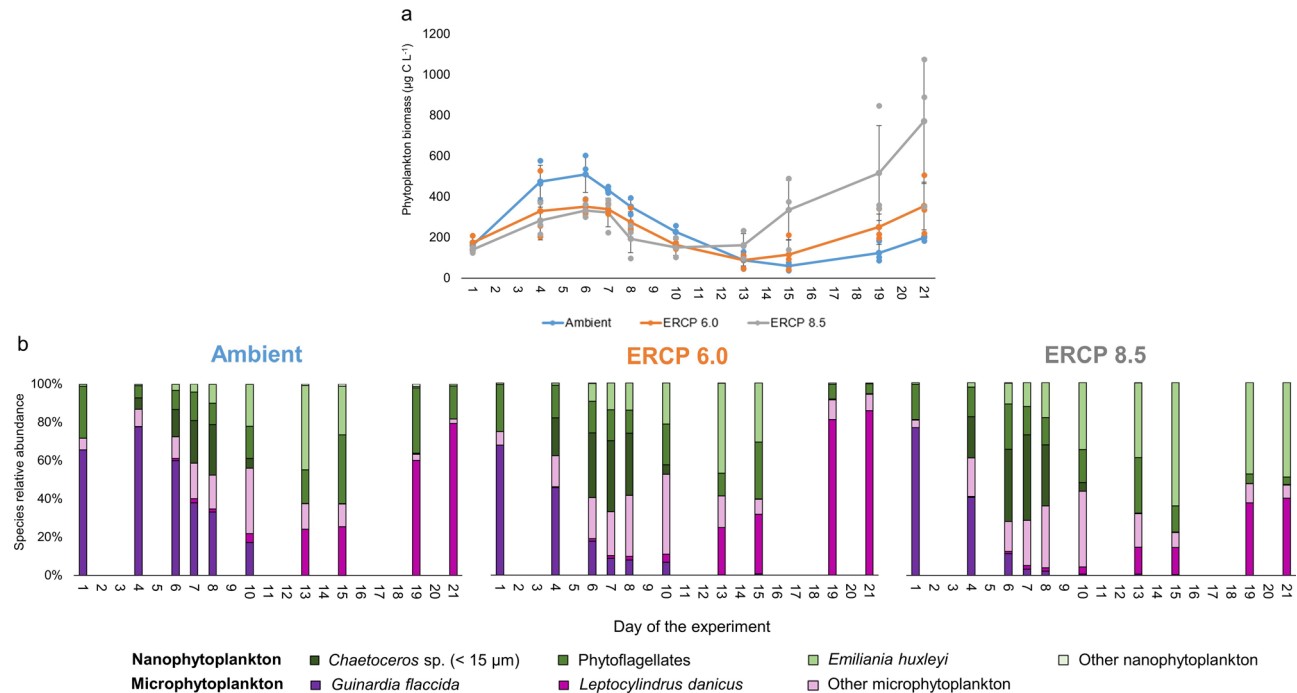

**Fig. 2 Phytoplankton biomass and community composition. a** Phytoplankton biomass; *x* axis represents the days of the experiment, different colours represent the Ambient treatment and extended representative concentration pathway (ERCP) scenarios (blue = Ambient, orange = ERCP 6.0, grey = ERCP 8.5), mean ± standard deviation. Cumulative phytoplankton biomass was not affected by the scenarios (LRT, df 86, $P = 0.46$, $n = 3$). **b** Relative abundances of different taxa of the phytoplankton communities.

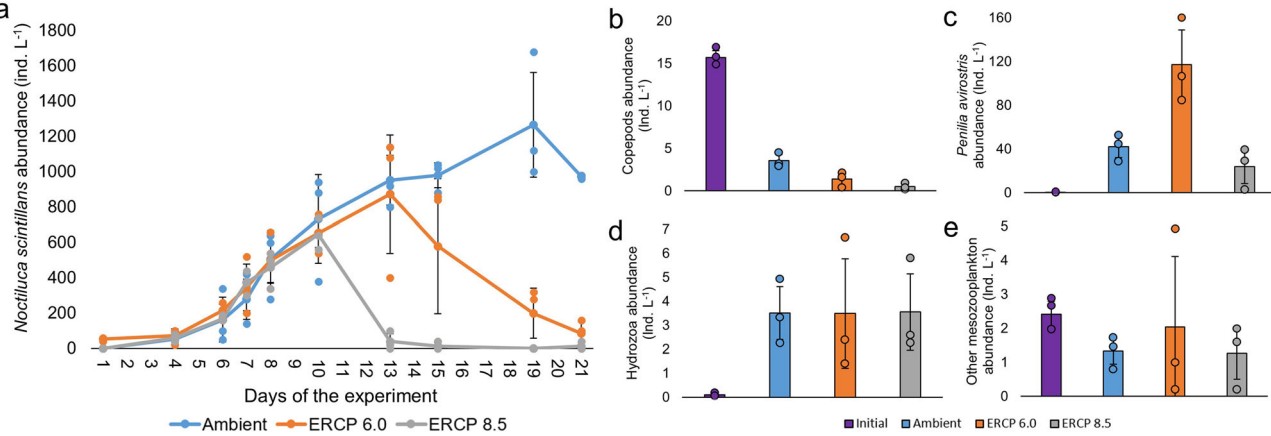

**Fig. 3 Mesozooplankton abundance and composition. a** Abundance of the dominant mesozooplankton species, *Noctiluca scintillans*, throughout the experiment period; *x* axis represents the days of the experiment, different colours represent the Ambient treatment and Extended Representative Concentration Pathway (ERCP) scenarios (blue = Ambient, orange = ERCP 6.0, grey = ERCP 8.5), mean ± standard deviation. *Noctiluca scintillans* abundance was significantly different across scenarios (LRT, df 86, $P < 0.0001$, $n = 3$). **b** Abundance of copepods. Copepods abundance is lower in all scenarios compared to Initial, but it is significantly higher in the Ambient compared to the ERCP scenarios (ANOVA, $F_{3,9}$ 276.1, $P < 0.0001$, $n = 3$). **c** Abundance of *Penilia avirostris*. The abundance of the cladoceran *Penilia avirostris* is higher in all scenarios compared to Initial (ANOVA, $F_{3,9}$ 26.62, $P = 0.0003$, $n = 3$). **d** Abundance of Hydrozoa. **e** Abundance of other mesozooplankton. Initial corresponds to values in situ when seawater for the experiment was collected. Ambient treatment and ERCP scenarios represent samples from day 15. Data are displayed as mean ± standard deviation.

For instance[15], used a semi-continuous microcosm approach to disentangle the direct temperature-mediated effects from indirect nutrient-limitation effects on phytoplankton size, and identified that nutrient effects largely dominate over direct temperature effects. While nutrient limitation has been associated with a reduction in light absorption leading to a reduction in cell size[17], small cells have low surface:volume ratios, which facilitates nutrient uptake efficiency and is therefore an advantageous feature in low nutrient waters[18]. In contrast to the two future scenarios, DIN was depleted before DIP in the Ambient scenario (Supplementary Fig. 2). These results are associated with our manipulation of N:P ratios, which are expected to increase in coastal seas[2], and support predictions that human-induced nitrogen enrichment is altering the balance with P[19]. Since the phytoplankton bloom rapidly depleted DIP in the ERCP 6.0 and 8.5 scenarios (Supplementary Fig. 2), we pose that the above-described phytoplankton biomass responses were mostly driven by DIP availability. The ERCP scenarios-induced smaller phytoplankton cell sizes are favourable for microzooplankton and as a consequence direct the flow of energy to the microbial food web, rather than efficiently fuelling higher trophic levels[20,21].

Following the bloom decay phase, abundances of the small coccolithophore *Emiliania huxleyi* increased in all treatments, but *E. huxleyi* only remained dominant in the ERCP 8.5 scenario until the end of the experiment, forming, together with the diatom *Leptocylindrus danicus*, a second phytoplankton bloom (Fig. 2a, b). The coccolithophore *E. huxleyi* has been the "canary" for ocean acidification research for a long time, as lower pH values are predicted to be detrimental to calcification processes present in this species[22]. Recent studies, however, challenge this view, showing the strain-specific response of this species to higher $pCO_2$[22]. In fact, it has been suggested that this phytoplankton species may become more competitive at higher $CO_2$ concentrations due to increased carbon fixing enzymatic activity[23,24]. Coccolithophore blooms, which are common during summer or early autumn in temperate regions[25,26], have increased in intensity over the past decades in the North Atlantic[27]. Furthermore, *E. huxleyi* has been reported to outcompete diatom blooms when nutrients, such as silica and phosphorus, become depleted[28–30]. While the calcification process

in *E. huxleyi* under high $pCO_2$ is modulated by temperature[31], positive effects of warming coupled with high $pCO_2$ on calcification of this coccolithophore have been reported[32]. This fact along with lower P availability may have created favourable growth conditions in the ERCP 8.5 scenario. Hence, we suggest that simultaneous $pCO_2$ and temperature increases, and lower dissolved nutrient concentrations, may promote intense *E. huxleyi* blooms in the future, which would significantly influence the role of this calcifying species in the marine carbon pump[12,33].

**The fate of larger grazers.** We observed a significant difference in the abundance of large grazers, from high in Ambient, to lower in ERCP 6.0, and even lower in ERCP 8.5 (Fig. 1, GLM, df 86, ERCP 6.0 $P = 0.36$, ERCP 8.5 $P = 0.0003$ and LRT, df 86, $P < 0.0001$, $n = 3$). The mesozooplankton community was largely dominated by the sea sparkle *Noctiluca scintillans*. Its abundance continuously increased from the beginning to the end of the experiment in the Ambient treatment (Fig. 3a), whereas this species died out on days 13 and 21 in the ERCP 8.5 and ERCP 6.0 scenarios, respectively (Fig. 3a). The abundance of copepods decreased during the experiment and was lower in both ERCP scenarios compared to the Ambient treatment (ANOVA, $F_{3,9}$ 276.1, $P < 0.0001$, $n = 3$, Fig. 3b). The second most numerous mesozooplankton species, the cladoceran *Penilia avirostris*, was more numerous on day 15 compared to initial values and was present in higher abundances in the ERCP 6.0 scenario and in lower abundances in the ERCP 8.5 scenario and Ambient treatment (ANOVA, $F_{3,9}$ 26.62, $P = 0.0003$, $n = 3$ Fig. 3b). This difference might result from an interaction between food availability, and nutritional requirements at elevated temperature and $pCO_2$. While temperatures during our experiment were well within the tolerance range of *N. scintillans*[34] and *P. avirostris*[35], this driver generally increases metabolic processes and energetic demands[36], and may intensify the sensitivity of consumers to low food availability. The scarcity of prey in the ERCP 8.5 might also have been the reason for the hump-shaped response of *P. avirostris* to the ERCP scenarios, as this species is not expected to be negatively affected by the temperature ranges used in our experiment. Given the correlation between temperature and metabolic rates, global warming could

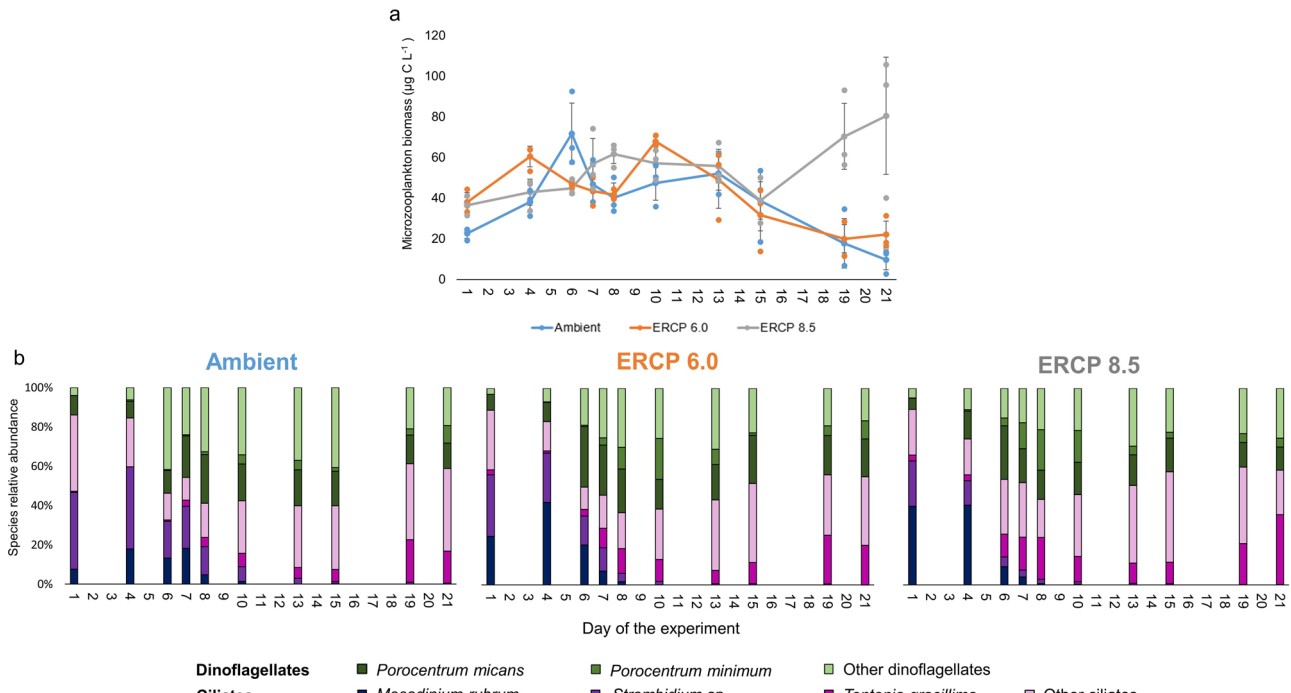

**Fig. 4 Microzooplankton biomass and community composition. a** Microzooplankton biomass; x axis represents the days of the experiment, different colours represent the Ambient treatment and extended representative concentration pathway (ERCP) scenarios (blue = Ambient, orange = ERCP 6.0, grey = ERCP 8.5), mean ± standard deviation. Microzooplankton biomass was significantly higher in the ERCP 8.5 (LRT, df 86, $P < 0.0001$, $n = 3$), compared to Ambient and ERCP 6.0 scenario. **b** Relative abundances of different taxa of the microzooplankton communities.

modify the metabolic demands of consumers, which, together with resource quality shifts, creates the potential for nutritional mismatches[37]. Recent work shows that the nutritional requirements of zooplankton, and the resource quality which maximises the growth of these ectotherms, is not constant but rather varies with temperature[38,39]. However, as seston C:N:P stoichiometry did not vary across treatments (Supplementary Fig. 3), bottom-up effects were likely driven by resource availability rather than by elemental stoichiometric quality. *Noctiluca scintillans* and *Penilia avirostris* can feed on a broad range of prey sizes[40–42], and may have been little affected by the shift in size from micro- to nanophytoplankton. Rather, we suggest that the lower phytoplankton biomass, and hence food availability, in the ERCP 6.0 and 8.5 scenarios, during the first bloom and its decay phase, were responsible for the differences observed. However, as there was no top-down control on meso-zooplankton during the experiment, it is important to note that the effects seen here could differ from communities in which their predators are present. In functional and numerical response experiments in which different phytoplankton taxa were fed to *N. scintillans*[43], identified, in addition to the importance of nutrient availability that this large heterotrophic dinoflagellate grew fast when fed with diatoms. Hence, the collapse of *N. scintillans* may be driven by a marginally non-significant increase from Ambient, to ERCP 6.0, to ERCP 8.5, in the proportion of diatoms within the phytoplankton community (LRT, df 86, $P = 0.05$, $n = 3$). Altogether, we suggest that multiple global change stressors may act synergistically and reduce the abundance of mesozooplankton in the future via altered food availability and demand, with potential consequences for higher trophic levels[44–47]. In parallel to bottom-up effects and to a lesser extent, we expect that the lower grazing pressure from meszoo-plankton might also have contributed to the increase of *Emiliania huxleyi* in the ERCP 8.5 scenario[48].

**Microzooplankton and the microbial loop**. The scenarios we tested had the opposite effect on microzooplankton than on

mesozooplankton. We observed a gradual increase in the biomass of microzooplankton from Ambient, to ERCP 6.0, to ERCP 8.5 scenarios (Fig. 1). Microzooplankton biomass increased along the first phytoplankton bloom and decreased after the phyto-plankton bloom had decayed (Fig. 4a). Whereas the micro-zooplankton biomass was not statistically different and continuously decreased until the end of the experiment in the Ambient and ERCP 6.0 treatments (GLM, df 86, $P = 0.16$, $n = 3$), the bloom of small coccolithophores in the ERCP 8.5 scenario coincided with an increase in microzooplankton biomass towards the end of the experiment (GLM, df 86, $P = 0.0004$, $n = 3$). Interestingly, coccoliths have been suggested as an effective defence mechanism against grazing from zooplankton[49], but a recent meta-analysis of data collected during mesocosm studies demonstrated that calcification of *E. huxleyi*, fails to deter microzooplankton grazing, thereby indicating that the possession of calcium carbonate scales does not provide *E. huxleyi* effective protection from microzooplankton grazing[50]. Moreover, bacterial biomass fluctuated during the experiment, it was higher at the end than at the beginning of the experiment in all treatments, and it reached higher levels in the ERCP 8.5 scenario than in the other two treatments during the decay phase of the first bloom (LRT, df 86, $P = 0.02$, $n = 3$, Supplementary Fig. 4). The increase in microzooplankton biomass at the end of the experiment might also be related to the increasing bacterioplankton biomass during this time, as picoplankton also provides an important source of food for these small grazers[51]. Together with the collapse of mesozooplankton in the ERCP 8.5 scenario, these results indicate that marine coastal planktonic food webs may shift from being mesozooplankton-dominated towards a dominant role of the microbial loop in response to global change in (Fig. 1). In support of this hypothesis, previous studies indicated that micro-zooplankton communities are rather unaffected by high $pCO_2$[52,53], and that the combination of warming and ocean acidification may in fact increase the interaction strength between

microzooplankton and their phytoplanktonic as well as bacterial prey[54–57]. Such shifts in bottom-up and top-down processes are not ecologically insignificant[58,59]. While microzooplankton are a natural trophic link between phytoplankton and bacteria, on the one hand, and mesozooplankton on the other hand[60], intensified trophic pathways through microzooplankton may diminish energy transfer efficiency to higher trophic levels. Strengthened energy flow through an additional trophic level leads to additional loss of organic carbon and, therefore, less efficient energy transfer to larger grazers[61,62]. The gain in prominence of microzooplankton over mesozooplankton we report here is supported by[63] and[62] who predicted lower energy transfer to higher trophic levels when the direct link from phytoplankton to mesozooplankton is shunted through an intermediary trophic level comprised of microzooplankton. Indeed, microzooplankton can directly compete with mesozooplankton for phytoplankton prey[64], and the addition of a trophic step between phytoplankton and mesozooplankton could reduce food web trophic efficiency, thereby creating a 'trophic sink' for production in the food web[65–67].

**Conclusions**. Here, we applied an integrated multiple driver design in a mesocosm experiment, to test the short-term effect of different global change scenarios on natural coastal plankton communities. This study identifies an ecological tipping point between the ERCP 6.0 and the ERCP 8.5 scenarios (Fig. 1). By promoting the growth of microzooplankton and nanophytoplankton, and by negatively impacting mesozooplankton, environmental conditions in the ERCP 8.5 scenario have the potential to considerably alter the structure and functioning of planktonic food webs in temperate coastal systems. In addition to these large structural shifts, we also observed that global change scenarios can cause the rise and demise of key species, such as *Emiliania huxleyi* and *Noctiluca scintillans*. The fact that planktonic food webs were relatively similar under Ambient and ERCP 6.0 conditions reinforces the goals of the 'Special Report on the impacts of global warming of 1.5 °C above pre-industrial levels'[68] to substantially reduce environmental risks and impacts of climate change.

## Methods

**Experimental design**. With an integrated multiple driver approach, we tested the influence of two global change scenarios on the structure and dynamics of plankton food webs based on predictions by the Intergovernmental Panel on Climate Change for the end of the 21st century[1]. Temperature and $pCO_2$ levels were chosen to represent (1) ambient conditions, (2) a moderate global change scenario based on RCP 6.0 (+1.5 °C and −0.2 pH) and (3) a more severe global change scenario based on RCP 8.5 (+3 °C and −0.3 pH). As nutrient inputs are also predicted to change considerably towards higher nitrogen to phosphorus ratio (N:P) in coastal European seas[2], we extended the RCP scenarios (ERCP) to include the predicted changing nutrient regime, with the Ambient and the ERCP scenarios having an N:P ratio (molar) of 16 (Redfield ratio) and 25, respectively, at the onset of the experiment.

**Mesocosm system**. The experiment was conducted in the mesocosm facility located at the Alfred-Wegener-Institut, Helmholtz-Zentrum für Polar- und Meeresforschung (AWI) Wadden Sea Station on the Island of Sylt[69]. The outdoor facility consists of 12 double-hulled, insulated, cylindrical tanks, made of UV stabilised high-density polyethylene (HDPE; Spranger Kunststoffe, Plauen, Germany). Each tank has a height of 85 cm, an inner diameter of 170 cm, and a net volume of 1800 L. To avoid the introduction of unwanted material, each mesocosm tank is covered with a translucent lid made of HDPE, which allows penetration of 90% of the photosynthetically active radiation. An adjustable flow-through system from the AWI Wadden Sea Station constantly supplies the tanks with fresh, unfiltered seawater. The temperature is regulated every 30 min by a Labview-based computer software (4H-Jena engineering, Jena, Germany), which periodically receives temperature data from Hydrolab DS5X Probes (OTT Messtechnik GmbH, Kempten, Germany) and controls external cooling units (Titan 2000 or Titan 4000 Aqua Medic, Bissendorf, Germany) and heaters (Titanium heater 500 W, Aqua Medic, Bissendorf, Germany).

We installed 450 L low-density polyethylene (LDPE) bags in each mesocosm tank (Supplementary Fig. 5). The LDPE was chosen as the material for the bags as it should not represent a risk for planktonic organisms either, since it also is used in

food industry for packaging. The LDPE bags were filled with seawater collected from the open North Sea with natural plankton communities (see details about filling procedures below). The bags were fixed in the centre of the tanks. By regulating the temperature and aerating the surrounding flow-through water as described above, we indirectly regulated temperature and $pCO_2$ in the LDPE bags. We replicated each treatment four times, for a total of 12 mesocosms. Due to damage to the bags and potential contamination of the plankton communities by the surrounding water, we excluded one replicate from each treatment, leaving triplicates for each of the three treatments. Despite the low number of replicates, the consistent response across scenarios and strong statistical results still reinforce the reliability of our results. The temperature in the Ambient conditions mesocosms was adjusted daily to the seawater temperature measured at the Helgoland Roads station (54°11.3'N, 7°54.0'E) and was increased by 1.5 and 3.0 °C for the ERCP 6.0 and ERCP 8.5 scenarios, respectively. We mounted small mortar mixer engines (TC-MX 1400-2 E, Einhell Germany AG, Landau/Isar, Germany) on top of each mesocosm tank, which were connected to a custom-made HDPE propeller (AWI, Helgoland, Germany). To avoid sedimentation of the planktonic organisms and mimic the relatively well-mixed water column condition found in the southern part of the North Sea[70], the submerged propellers gently homogenised the water column of the LDPE bags at 50 rpm in a 1-min-mixing/30-min-pause interval. To reach the desired $pCO_2$ in the different ERCP scenarios, streaming pipes aerated each tank with the desired gas mixture in the water outside the LDPE bags (Supplementary Fig. 5). The aeration outside the mesocosm bag was intended to prevent damage to fragile planktonic organisms that are sensitive to bubbling. The Ambient conditions mesocosms were bubbled with pressured air, ERCP 6.0 scenario with 800 µatm $pCO_2$ and the ERCP 8.5 with 1000 µatm $pCO_2$, which were determined by a central $CO_2$-mixing facility (GMZ 750, HTK, Hamburg, Germany). The mesocosm cover trapped the $pCO_2$-controlled atmosphere above the mesocosm water column, hence realistically mimicking future environmental conditions.

**Seawater collection and filling of the mesocosm bags**. On August 14, 2018, we collected water from the open North Sea 45 km west of the island of Sylt (55°01'20.0"N 7°38'41.0"E), during a cruise with the AWI research vessel Uthörn. During the water collection and filling procedure of the mesocosm bags, we did not use any pumps, but transferred seawater via gravity flow to prevent any damage to fragile organisms within the planktonic community. To sample seawater onboard, we submerged a 500 L tub attached to a crane to fill it with seawater from the upper 5 m sea surface. The tub was subsequently lifted up to let the water flow through a hose connected to the tub into 1000 L polyethylene Intermediate Bulk Containers (IBC, AUER Packaging GmbH, Amerang, Germany). We attached a 1000-µm mesh to the end of the hose, to exclude larger organisms, such as jellyfish and fish larvae. This procedure prevented any disproportionally large impact which larger consumers can have on the rest of the plankton community in a 450 L enclosed water volume. Furthermore, this approach enabled us to focus on bottom-up processes since there was no top-down control on mesozooplankton. The procedure was repeated until eight IBC tanks were filled (8000 L), which took about 3 h.

Before filling the mesocosm bags, we first gently homogenised the water in the IBC tanks. Then, we attached a four-way distributor to one IBC tank, and the tank was lifted by a wheel-loader to allow gravity flow of the seawater into the mesocosm bags. At the end of each connected hose, a flowmeter measured the exact volume of water that was released into each mesocosm bag. We filled 80 L of seawater simultaneously to four bags, and then filled the next quadruplet of mesocosms. This enabled an equal distribution of the water contained in each IBC tank among the twelve mesocosms. This procedure was repeated until all mesocosm bags were filled with 450 L of North Sea seawater. This procedure enabled us to successfully tackle a major challenge when conducting mesocosm experiments, the difficulty of achieving homogenous replicates at the onset of the experiment[10]. Across scenarios, no significant difference was found in biomass of phytoplankton, microzooplankton and bacterioplankton on the first day of the experiment (Kruskal–Wallis test, df = 2, $P > 0.05$). Once the filling procedure was completed, we directly measured the dissolved N and P concentrations in each mesocosm bag according to the method described in ref. [71], and subsequently adjusted the dissolved N:P ratios to 16 (Ambient conditions) and 25 (ERCP scenarios). We added DIN to reach 5 µmol $L^{-1}$ in all scenarios, DIP to reach 0.31 µmol $L^{-1}$ in the Ambient scenario and 0.2 µmol DIP $L^{-1}$ in the ERCP 6.0 and 8.5 scenarios. These values correspond to mean values for that period of the year according to data from the Helgoland roads time series. At the onset of the experiment, we bubbled a small volume of seawater with pure $CO_2$, which lowered its pH to 4.8 at saturation. Using a 50 mL plastic syringe connected to a 1-m hose, we injected 400 mL (ERCP 6.0) and 760 mL (ERCP 8.5) of the saturated $CO_2$ seawater at the bottom of the mesocosm bags to reduce the initial pH values by −0.2 and −0.3 for the ERCP 6.0 and ERCP 8.5 scenarios, respectively. During the rest of the experiment, the pH was influenced by the planktonic communities through photosynthesis and respiration, and by the atmospheric $pCO_2$ (see above).

**Physical-chemical conditions in the mesocosm bags**. Temperature, pH, light irradiance and salinity were measured every day at 9:00 (Supplementary Fig. 6). Light intensity was measured just below the water surface with a Li-cor Li-250 Light metre (Bad Homburg, Germany). Temperature measurements were done directly inside the mesocosm bags using a Testo 110—temperature metre

(Lenzkirch, Germany). Total alkalinity (TA) samples were taken by plunging, filling, and closing an air-tighten 100 mL transparent glass bottle inside the mesocosm to avoid air bubbles. The samples were stored at 4 °C before being analysed within 36 h through linear Gran-titration[72] using a TitroLine alpha plus (Schott, Mainz, Germany). Samples for dissolved inorganic nutrients and TA were taken at an interval of 1–3 days depending on the phytoplankton bloom development.

For further analyses, water was collected from each mesocosm bag with clean plastic beakers and brought to the lab for processing. The first parameter measured was pH using a WTW pH 330i equipped with a SenTix 81 pH electrode (Letchworth, England). Salinity was measured with a WTW CellOx 325 (probe Oxi 197-S, Letchworth, England). Dissolved inorganic nutrient samples were collected with a sterile plastic syringe and filtered through a 0.45 µm PTFE filter (Minisart, Sartorius, Goettingen, Germany) fitted to the syringe. For this step, the first 2 mL of the sample were used to rinse the filter and directly discarded. Samples for dissolved inorganic nitrogen (DIN) and phosphorus (DIP) were stored at −20 °C, and the samples for dissolved silica (DSi) were stored at 4 °C, until photometric analyses[71] (Supplementary Fig. 2). Results of TA, pH, temperature, salinity, atmospheric pressure, DIP and DSi were computed to determine the carbonate system using the CO2Sys Excel Macro[73] with a set of constants defined by[74] (Supplementary Data 1). Although $pCO_2$ in the mesocosms were below levels projected by the RCP scenarios during the experiment, $CO_2$ concentrations were always different across scenarios within the expected gradient (Supplementary Fig. 6b and Supplementary Data 1), where Ambient is lower than ERCP 6.0 that is lower than ERCP 8.5. Given the extreme complexity of keeping $pCO_2$ constant in mesocosm experiments, even with an appropriate $CO_2$ atmosphere, and especially throughout a phytoplankton bloom event that is able to change dissolved $CO_2$ even in the open sea[75]. Therefore, our approach yields the most realistic of $CO_2$ time courses in a future ocean. The remaining water was used for analyses of the planktonic community.

**Planktonic community.** To determine plankton species composition and biomass, 100 mL of mesocosm seawater were stored in amber glass bottles and immediately fixed with neutral Lugol's iodine solution (1% final concentration) to preserve calcifying phytoplankton. Another 250 mL were fixed with acid Lugol's iodine solution (2% final concentration) to preserve other phytoplankton and microzooplankton species. Phytoplankton were identified using an inverted microscope Zeiss Anxiovert 135 (Jena, Germany) and microscope using a Zeiss Axio Observer 7 A1 (New York, USA) following the method described in[76]. Due to the high biomass of the mesozooplankton *Noctiluca scintillans* during the experiment, this species was quantified and identified by the Uter-möhl method as well, using chamber volumes ranging between 50 and 100 mL. Planktonic organisms were identified to species level, or pooled into size-shape dependent groups when species identification was not possible. Scanning electron microscopy (Philips XL30 SEM, Massachusetts, USA) was applied to identify coccolithophore species by the morphology of the coccoliths. For this procedure, prior to microscopy, 5 mL of the neutral lugol fixed sample were filtered through a 0.2 µm pore size polycarbonate membrane filter (Merck Millipore, Burlington, USA), dried in a drying oven at 40 °C for 12 h, placed on a metal stub using an adherent carbon disc with increased conductivity, and then sputter-coated with a 10-nm gold layer.

Mesozooplankton, with the exception of *Noctiluca scintillans*, was sampled with a plankton net (200 µm) in situ (Initial) when seawater was collected, and on the day 15 of the experiment by sieving 5 L of seawater from the mesocosm through a 200-µm nylon mesh. The organisms caught on the mesh were flushed back into a 50 mL transparent Kautex container with sterile filtered seawater (0.2 µm), and immediately fixed with formaldehyde. The mesozooplankton community composition was determined by counting the whole sample or three subsamples when splitting was necessary with a Folsom splitter[77,78]. The counting took place using a Bogorov chamber under stereomicroscope (Leica M205), and taxonomic identification was conducted as in ref. [79]. Samples for bacterioplankton biomass were taken as 5 mL of seawater, sieved through 20-µm nylon mesh, fixed with glutaraldehyde (0.1% final concentration) and frozen at −80 °C until analysis. The samples were thawed in water bath (20 °C) and stained with SYBR Green (Invitrogen) following the method described by Marie et al.[80]. Bacteria cells were enumerated by flow cytometry (BD AccuriTM C6 Plus, BD Biosciences) with a flow rate of 12 µL min$^{-1}$ for 1–2 min and diluted in sterile filtered seawater (0.2 µm) when bacterial cell number was higher than 400 events s$^{-1}$. As SYBR Green stains DNA molecules without distinguishing taxonomical groups, our results of bacterioplankton include any organisms within the range of picoplankton cell size (~0.2–2 µm), including picocyanobacteria.

Biovolume of each phytoplankton and microzooplankton species was calculated from the measurement of cell dimensions using geometric formulae according to ref. [81]. Cell volume was converted into carbon following the equations of[82] for diatoms (pg C cell$^{-1}$ = 0.288 × V$^{0.811}$), dinoflagellates (pg C cell$^{-1}$ = 0.760 × V$^{0.819}$) and other protist plankton with the exception of ciliates (pg C cell$^{-1}$ = 0.216 × V$^{0.939}$), where V is the cell volume in µm$^3$. Ciliate carbon content was calculated as 0.19 pg C µm$^{-3}$ according to ref. [83]. *Noctiluca scintillans* C content was determined as 0.138 µg C cell$^{-1}$[84]. Bacteria cell counts were converted into carbon using the 20 fg C cell$^{-1}$ factor defined by Lee and Fuhrman[85]. The box size on the infographic of biomass (Fig. 1) was determined by

the integral area under the curve of the plankton biomass over time (Figs. 2a, 3a, 4a and Supplementary Fig. 4) and dominant taxa followed the values of the relative abundance of the most abundant taxa (Figs. 2b, 3b and 4a, b). Elemental composition (CNP) of seston was determined by filtering 200 mL of seawater through precombusted GF/F filters. Carbon and nitrogen content were measured with a Vario Micro Cube elemental analyser (Elementar, Hanau, Germany). Phosphorus content was quantified as orthophosphate after oxidation by molybdate-antimony[70]. Functional groups were determined as Phytoplankton, Bacterioplankton, Microzooplankton and Mesozooplankton. The phytoplankton group included diatoms, phytoflagellates and autotrophic dinoflagellates, according to the descriptions of trophic mode for each species[86]. The microzooplankton group comprised heterotrophic and mixotrophic dinoflagellates and ciliates, including nanociliates (< 20 µm). Mesozooplankton species were all the heterotrophic organisms larger than 200 µm.

**Statistics and reproducibility.** Statistical analyses were performed using R 3.4.3 software[87]. For all analyses, the threshold of significance was set at 0.05. All statistical analyses were applied considering the three individual replicates per scenario. Each replicate was determined as one tank of the mesocosm system. The effect of the ERCP scenario on planktonic biomass was assessed by a generalised linear model (GLM). We first fitted a model of total biomass (either phytoplankton or zooplankton) depending on treatments. It allowed us to check for general treatment effect on planktonic biomass. Then, we created a second model including treatment and time. By comparing the constrained model (time and scenario) against the unconstrained one (only scenario) by Likelihood ratio test (LRT), we could test whether timing in planktonic biomasses were similar among treatments. Effects of the ERCP scenarios on the phyto- and microzooplankton species composition and affinity of species to the scenarios were analysed through the principal response curve (PRC) using the 'vegan' R package. This test shows the degree of difference over time of the community composition in the ERCP scenarios in comparison to the Ambient condition, which is set as a control (effect '0'). Species weights are analysed as means of their regression coefficient against the control. When the curve of difference of the ERCP scenario has a positive slope, positive values for species weights represent affinity of this species to the scenario, whereas negative values would represent the negative effect of the scenario on such species and vice versa. Differences of mesozooplankton abundance were analysed through Analysis of Variance (ANOVA) followed by a post hoc test (Tukey test). Data were log-transformed when normality and homoscedasticity of residuals were not met for ANOVA and LRT.

**Reporting summary.** Further information on research design is available in the Nature Research Reporting Summary linked to this article.

## Data availability
The datasets generated and analysed during this study are available in the Pangaea repository: https://doi.pangaea.de/10.1594/PANGAEA.940529.

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

## Acknowledgements
H.D.M., M.K. and C.L.M. were supported by the German Federal Ministry of Education and Research (BMBF grant no. 01LN1702A). M.B. was funded by the German Science Foundation (DFG) with the Priority Program DynaTrait. We thank the colleagues from Alfred-Wegener-Institut for the technical and scientific support during the experiment, especially Julia Haafke, Petra Kadel, Silvia Peters, Andreas Kornmann, Inga Kirstein, Johannes Rick, Ragnhild Asmus and Harald Asmus. Sincere thanks to the colleagues who supported us in analysing some of the samples, including Ursula Ecker (mesozooplankton), Tatyana Romanova (dissolved inorganic nutrients), Bernhard Fuchs (bacterioplankton), Sebastian Rokitta and Gernot Nehrke (scanning electron microscopy). We thank Herwig Stibor, Helmut Hillebrand and Ulf Riebesell for providing their expert opinion on the experimental design. We thank Herwig Stibor and Maria Stockenreiter for providing the LDPE bags in which the experiment was conducted.

## Author contributions
Conceptualisation: H.D.M., K.H.W., M.B. and C.L.M. Data acquisition: H.D.M., M.K., N.T. and C.L.M. Data analysis: H.D.M. and J.D.P. Writing—original draft: H.D.M. and C.L.M. Writing—review and editing: H.D.M., M.K., J.D.P., N.T., K.H.W., M.B. and C.L.M.

## Funding

## Competing interests
The authors declare no competing interests.
