## [Peer Review File · Communications Biology]

Reviewers' comments:

Reviewer #1 (Remarks to the Author):

This study describes multi-trophic level mesocosm experiment that simulates warming, high CO₂ concentration and nutrient ratio manipulations. The results are realistic, convincing and interesting. One of the major findings is the enhancement of microzooplankton and the microbial loop. The only weakness on this study is the lack of data of some key components of the microbial food web. For example picocyanobacteria (e.g. *Synechococcus*) are key components of the phytoplankton and its abundance is likely increasing in this system (Schmidt et al. Increasing picocyanobacteria success in shelf waters contributes to long-term food web degradation. *Glob Change Biol.* 2020; 26: 5574– 5587. <https://doi.org/10.1111/gcb.15161>), with consequences on the food web (Brander, K. and Kjørboe, T. (2020), Decreasing phytoplankton size adversely affects ocean food chains. *Glob. Change Biol.*, 26: 5356-5357. <https://doi.org/10.1111/gcb.15216>). Pico-phytoplankton could be accessed by flow cytometry. Why this data does not appear in the manuscript? Was it analyzed? Other microbial food web compartments were not shown such as nanoflagellates and small ciliates that can only be quantified by epifluorescence microscopy. Bacterioplankton, the only component of the microbial food web shown in the manuscript) is an important component strongly regulated by nutrients and predation (metazoans) but its abundance remains relatively constant (Gasol et al. 2002. Regulation of bacterial assemblages in oligotrophic plankton systems: results from experimental and empirical approaches. *Antonie Van Leeuwenhoek* 81, 435–452. <https://doi.org/10.1023/A:1020578418898>). Herbivory is probably a major carbon pathway, more than is bacterivory (Sherr EB, Sherr BF. Bacterivory and herbivory: Key roles of phagotrophic protists in pelagic food webs. *Microb Ecol.* 1994 Sep;28(2):223-35. doi: 10.1007/BF00166812. PMID: 24186449.) . Herbivores include small flagellates and small ciliates, beside heterotrophic dinoflagellates in the microzooplankton, and most of these can not be quantified properly in lugol fixed samples. Therefore I think the manuscript missed some key components of the microbial food web, which probably will become more important with climate change.

Reviewer #2 (Remarks to the Author):

The manuscript by Moreno et al. presents the results of a short-term mesocosm experiment, where warming, acidification, and nutrient enrichment were simultaneously manipulated in three treatments: an ambient control, and two treatments where they were elevated in line with two IPCC scenarios for the end of the century. The authors found that the strongest IPCC scenario led to an overall reduction in the biomass of meso-zooplankton and micro-phytoplankton, and an increase in the biomass of micro-zooplankton and nano-phytoplankton, i.e. apparent cascading effects through the planktonic portion of their model marine food webs. The greater similarity of the intermediate IPCC scenario to the ambient control indicates that there may be a tipping point at which these dramatic changes occur.

I generally enjoyed reading the manuscript, although I found the discussion to be quite speculative in places and I have some concerns that may dull the impact of the work. While it is admiral to map the experimental manipulations onto the IPCC scenarios, the simultaneous manipulation of three stressors means we cannot determine which of the stressors are causing which effects...it might be one, it might be all. This leads to difficulties in the interpretation of the results, where the authors often relate findings to previous studies on individuals stressors or a combination of two of their stressors, but we can't really know what the underlying drivers are here.

Furthermore, there is some discussion about the consequences for higher trophic levels, but that is also quite speculative when they were not in fact manipulated here. Within these mesocosms, meso-zooplankton are the apex predators, and they are not exerted to any top-down control from larger plankton or fish species. The results could be completely different if we add those higher trophic levels into the equation, and indeed there may be some quite complex interactions between bottom-up and top-down control. This is just one example of why the authors should be very careful not to over-interpret their findings when relating them to real systems.

The experiment is short, at just 21 days, which is probably sufficient to cover the immediate

response of these small planktonic taxa to the treatments. But I was left wondering how well such a short-term exposure to warming/acidification maps onto a real climate change scenario, where the organisms have many months, years, or even decades to "get used" to the treatments. Seasonal or intergenerational responses could change the story quite a bit, particularly as different rates of acclimation and adaptation across different trophic levels alter the response of individual taxa.

In summary, I think this experiment presents some very interesting results that will be a valuable addition to the broader literature, but they do need to be carefully couched within the quite limited setting of the experimental conditions, duration, and trophic complexity that they were examined under.

Some specific comments:

Ln33. Change to either "greenhouse gas emissions" or "emissions of greenhouse gases"

Ln47. Change to either "copepod abundance" or "abundance of copepods"

Ln52. "Amiable" is not the right word here...change to something like "conducive"

Ln100. There is statistical support for the first part of this sentence (related to *Guinardia flaccida*), but no such support for the second part about contribution of nanophytoplankton. Are these differences between treatments significant? It could also be made clearer whether the ambient was significantly different from both ERCP 6.0 and 8.5, and whether the latter two were or were not significantly different from one another.

Ln119. This sentence is too vague. "Affecting" primary production how and what are some of the potential "subsequent food-web effects"?

Ln121. That's a reasonable assumption, but it is just speculation since you cannot disentangle the independent effects of nutrients, warming, and acidification in this experiment. Try to be clear about that throughout and not over-interpret the role of individual stressors in your results. It would also help to provide stronger support from the literature for the role of P-limitation in lowering phytoplankton size and overall biomass.

Ln130. The discussion from here until Ln148 is interesting, although I am left wondering why the species may be viewed as a "canary" if there is ample research showing that it can actually thrive in acidified conditions. Could this be due to the combined effects of warming and nutrients on acidification, as you briefly allude to at Ln148? I think we need to see a little more development of that discussion, and particularly the evidence for why the coccolithophore species shows such variable responses to acidification.

Ln154. Again, why present just one overall statistic and not the pairwise comparisons that are actually presented in the preceding text? I'm thinking about the LRT equivalent of a Tukey post-hoc test here to see which treatments were significantly different from one another.

Ln158. It's almost impossible to see the difference you describe here in Figure 3b. I suggest presenting separate barplots for each of the four taxa crammed into that figure, allowing you to put them each on sensible y-axis scales. It should be feasible to squeeze four small panels into the space currently occupied by Figure 3b, using a single y-axis label and colouring by treatment, rather than taxa for consistency with the other figures.

Ln164. I find the discussion from here until Ln193 to be quite speculative and unclear. Metabolism was not measured, nor was nutritional demand quantified, so speculating on the role of these processes in the prevalence of a particular species (*Penilia avirostris*) feels quite tenuous. It also doesn't quite make sense to me, when the treatment where it was most abundant had an intermediate temperature and nutrient availability to the other two treatments. This indicates a hump-shaped response, and yet you said that temperatures were "well within the tolerance range" of the species. Have a rethink about the message in this section because it doesn't quite flow.

Ln190. This is of course likely, but also quite speculative since you did not quantify the biomass of higher trophic levels in your experiment. A more complex trophic structure may have a whole host of interesting feedbacks (e.g. through altered top-down control) that could fundamentally change the conclusions of your experiment (which put a cap on the trophic pyramid at mesozooplankton, removing any role of top-down control on their biomass).

Ln196. Where is the statistical support for this statement? Note that it feels like a different statement to the subsequent two sentences, i.e. a comparison of treatments, versus statements about the trajectory of biomass through time within the various treatments. There is a general lack of transparency about the underlying statistics throughout the paper.

Ln218. I'm a little confused by the discussion here. If interaction strengths are expected to strengthen between microzooplankton and their bacterial prey, then why would bacterial biomass have increased in tandem with microzooplankton biomass in the ERCP 8.5 treatment, as indicated at Ln210? There are a lot of these little inconsistencies throughout the discussion that means the explanation of the results does not quite gel.

Ln220. I think "anodyne" may be the wrong choice of word here... "ecologically insignificant" perhaps?

Ln223. Why would intensified trophic pathways through microzooplankton diminish energy transfer efficiency? The logic is unclear and lacking support from the literature.

Ln274. It's actually pretty surprising to see that the mesocosms were lined with plastic bags. Are there any concerns about this artificial substrate (flexible plastic seems even less realistic than the typical hard, smooth sides of a mesocosm), or indeed that it may act as a fourth stressor in the experiment (i.e. plastic pollution has been shown to significantly alter marine communities)?

Ln280. I think it should be made clear in the main text that the experiment actually just consists of three replicates of each of the three treatments...this really is quite a skeletal design and likely gives you very little statistical power to detect differences between treatments. That puts quite a bit of uncertainty over the story that is presented.

Ln284. It occurs to me that you are effectively taking ambient communities and then suddenly exposing them to water that is 1.5 or 3.0 °C warmer than they are used to. Could the "shock" of this sudden change lead to many of the effects that you subsequently see, and how much does this differ from a real climate change scenario, which will slowly ramp up over many years?

Ln321. The procedure does indeed seem to argue for a good likelihood of homogenous starting communities...but was this tested through sampling? It would be really great to show no treatment effects in the "before" communities to dispel any concerns here, and could convert this into a full before-after-control-impact style experimental design, which would be more powerful.

Ln359. I don't want to make too big a deal of this, but it is a little concerning that the acidification treatment did not actually match the IPCC scenarios. Similarly, it seems like the nutrient concentrations were not all that different (even at the beginning of the experiment for the two ERCP scenarios). It might be worth acknowledging that in the main text, i.e. the goal was to imitate IPCC scenarios, but this was not perfectly realised in the experiment.

Ln431. The description of the statistical analysis here seems to indicate that the LRT only distinguishes a significant difference between the ambient control and the combined ERCP 6.0 and 8.5 scenarios. Is that the case, or have I misinterpreted the description? If the latter, it needs a better description...if the former, it seems like this is not a very good test for determining treatment differences, since we cannot tell the difference between the two ERCP scenarios. I did not get that impression from the main text, although I was frustrated by the lack of statistical clarity on which treatments were different from each other, so something needs to be improved here (either in the statistics or in the presentation and description).

Reviewer #3 (Remarks to the Author):

Dear authors,

I have read the following manuscript „Structural alterations of planktonic food webs under global change: an integrated multiple driver mesocosm experiment“ carefully and find it very well done and represents a good contribution to this field of knowledge. I liked the overall manuscript very much and my comments tend not to be detailed but rather a bit broader. While the data is nicely analysed and presented with the appropriate statistical tests, I only have some concerns about the methodology used. However, I think this can be discussed very easily by you. The water used in the mesocosms for the experiment was taken directly from the sea. My first question is from what depth was this taken? Was this a mixed sample from several depths or only water from the upper layers. This would be very interesting, especially as the depths of the mesocosms are quite limited. This brings me to the next comment. The mesocosms described here are shallow, and given the very great depth of the upper layer in the free ocean, this could have an influence on the course of the experiment and thus on the data collected. Specifically, I mean that a very complex community and its interrelationships were observed in a rather unusual part scenario. It would be very good if you could briefly address this issue. Because this setup represents another stress factor, that of lower mixing depth. Mesocosm experiments with shallow experimental depth can lead to particular turn over events. This should definitely be discussed in this manuscript,

especially as you relate the results onto future climate change scenarios. If this experimental problem were discussed, it would support the weighting of the results even more.

As you describe, there was a temperature control and temperature was regularly compared with the open sea and accordingly adjusted in approaches with increased temperature. However, I wonder if there was not also a temperature increase in the ambient treatments, as the mesocosm facility is land-based, and if so, how was this compensated for or included in the data analysis?

Furthermore, I think the manuscript is very well written and the data is certainly of broad interest to a wide readership.

Communications Biology
Revision

Hugo Duarte Moreno
Alfred Wegener Institute
Helmholtz Centre for Polar and Marine
Research
Ostkaje 1118, 27498
Helgoland, Germany
hugo.moreno@awi.de
tel: +49 172 2620219

November 11th 2021

Dear Reviewers,

Thank you for your interest in our manuscript. We have read all your comments and modified the manuscript accordingly. We hope the text is more consistent now.

Please find below our remarks and answers to your questions.

Reviewer #1 (Remarks to the Author):

This study describes multi-trophic level mesocosm experiment that simulates warming, high CO₂ concentration and nutrient ratio manipulations. The results are realistic, convincing and interesting. One of the major findings is the enhancement of microzooplankton and the microbial loop. The only weakness on this study is the lack of data of some key components of the microbial food web. For example picocyanobacteria (e.g. *Synechococcus*) are key components of the phytoplankton and its abundance is likely increasing in this system (Schmidt et al. Increasing picocyanobacteria success in shelf waters contributes to long-term food web degradation. *Glob Change Biol.* 2020; 26: 5574– 5587. <https://doi.org/10.1111/gcb.15161>), with consequences on the food web (Brander, K. and Kjørboe, T. (2020), Decreasing phytoplankton size adversely affects ocean food chains. *Glob. Change Biol.*, 26: 5356-5357. <https://doi.org/10.1111/gcb.15216>). Pico-phytoplankton could be accessed by flow cytometry. Why this data does not appear in the manuscript? Was it analyzed? Other microbial food web compartments were not shown such as nanoflagellates and small ciliates that can only be quantified by epifluorescence microscopy. Bacterioplankton, the only component of the microbial food web shown in the manuscript) is an important component strongly regulated by nutrients and predation (metazoans) but its abundance remains relatively constant (Gasol et al. 2002. Regulation of bacterial assemblages in oligotrophic plankton systems: results from experimental and empirical approaches. *Antonie Van Leeuwenhoek* 81, 435–452. <https://doi.org/10.1023/A:1020578418898>). Herbivory is probably a major carbon pathway, more than is bacterivory (Sherr EB, Sherr BF. Bacterivory and herbivory: Key roles of phagotrophic protists in pelagic food webs. *Microb Ecol.* 1994 Sep;28(2):223-35. doi: 10.1007/BF00166812. PMID: 24186449.) . Herbivores include small flagellates and small ciliates, beside heterotrophic dinoflagellates in the microzooplankton, and most of these can not be quantified properly in lugol fixed samples. Therefore I think the manuscript missed some key components of the microbial food web, which probably will become more important with climate change.

Answer:

This is a very valid argument, and we apologize for the lack of clarity in the description of our work, as we did include most of the groups mentioned by the reviewer in some way, but we did not obviously separate from other groups, however. We fully agree that micro- and nano grazers are very

important, and, as argued by the reviewer, likely to become more important in future. Picocyanobacteria was enumerated in our experiment together with other picoplankton species by flow cytometry. As SYBR Green stains DNA without distinguishing taxonomical groups, and our flow cytometry counts considered FSC (forward scatter) as a proxy for size, our results of bacterioplankton include organisms within the range of picoplankton cell size (~0.2 – 2 µm). Therefore, picocyanobacteria are included in our bacterioplankton counts, although it is not distinguished from heterotrophic prokaryotes. Nanoflagellates and nanociliates are also included in our counts of nanoplankton, and we did distinguish them from other groups. By applying a 400x magnification on the inverted microscope we used for taxonomic identification, we identified cells down to 2 µm. Due to their distinct cell shape, we can successfully identify nanoflagellates and small ciliates. In our study, we classified nanoflagellates as (nano)phytoflagellates and small ciliates were grouped together with other larger ciliates for the sake of clarity when presenting the data. Yet, we have individual counts of every group split in cell shape and size. We agree that the Uthermöhl method has limitations in quantifying nanoplankton, due to the extensive time necessary to search all small cells, but we would have noticed any effect of the scenario we tested on these small-sized groups, as we did with nanoflagellates. We have changed the text accordingly (Ln492), making sure that we explicitly state where we have categorized all of the groups mentioned by the reviewer, so that it becomes clear how and where these groups were dealt with. We also thank the reviewer for drawing our attention to previous studies, which we now refer to in our manuscript (Ln280).

Reviewer #2 (Remarks to the Author):

The manuscript by Moreno et al. presents the results of a short-term mesocosm experiment, where warming, acidification, and nutrient enrichment were simultaneously manipulated in three treatments: an ambient control, and two treatments where they were elevated in line with two IPCC scenarios for the end of the century. The authors found that the strongest IPCC scenario led to an overall reduction in the biomass of meso-zooplankton and micro-phytoplankton, and an increase in the biomass of micro-zooplankton and nano-phytoplankton, i.e. apparent cascading effects through the planktonic portion of their model marine food webs. The greater similarity of the intermediate IPCC scenario to the ambient control indicates that there may be a tipping point at which these dramatic changes occur.

I generally enjoyed reading the manuscript, although I found the discussion to be quite speculative in places and I have some concerns that may dull the impact of the work. While it is admirable to map the experimental manipulations onto the IPCC scenarios, the simultaneous manipulation of three stressors means we cannot determine which of the stressors are causing which effects...it might be one, it might be all. This leads to difficulties in the interpretation of the results, where the authors often relate findings to previous studies on individual stressors or a combination of two of their stressors, but we can't really know what the underlying drivers are here.

Answer:

We thank the reviewer for the generally positive appraisal of our work. The reviewer is correct that the scenario approach we employed cannot distinguish between the effect of each tested environmental driver separately. However, we would like to stress that this was not our aim. As changes in temperature, pCO₂, and N:P ratios are taking place simultaneously, the impact assessment of future conditions on planktonic communities requires a multiple-driver approach. Hence, it is currently of utmost importance to make accurate and reliable predictions of the fate of planktonic communities in future conditions, and we believe that our work provides a more realistic assessment of these drivers'

impact than an experiment addressing drivers singly would. We have added this information to guide the reader through our experimental approach (Ln 54). Despite the benefits of multiple-driver approaches, their complexity has led to a disparity in the literature, with a majority of studies focusing on single-driver experiments. Hence, the literature cited in our discussion reflects the information available, rather than a wish to identify the most influential driver(s).

Furthermore, there is some discussion about the consequences for higher trophic levels, but that is also quite speculative when they were not in fact manipulated here. Within these mesocosms, meso-zooplankton are the apex predators, and they are not exerted to any top-down control from larger plankton or fish species. The results could be completely different if we add those higher trophic levels into the equation, and indeed there may be some quite complex interactions between bottom-up and top-down control. This is just one example of why the authors should be very careful not to over-interpret their findings when relating them to real systems.

Answer:

The reviewer raises an important point here. When sampling the seawater for our experiment, we screened the water through a 1000 μm mesh, to exclude larger organisms, such as jellyfish and fish larvae. This procedure is commonly conducted in mesocosm experiments, as larger consumers can have a disproportionately large impact on the rest of the plankton community in an enclosed water volume. As pointed out by the reviewer, this procedure enabled us to focus on bottom-up processes since there was no top-down control on mesozooplankton. This is now explained in our manuscript (Ln 240). Since manipulating mortality regimes based on future predator densities would have been speculative as there is no real consensus of what is going to happen, we decided not to include this aspect. We followed the reviewer's recommendation, and modified the manuscript to avoid any speculation on the consequences for higher trophic levels (Ln 223).

The experiment is short, at just 21 days, which is probably sufficient to cover the immediate response of these small planktonic taxa to the treatments. But I was left wondering how well such a short-term exposure to warming/acidification maps onto a real climate change scenario, where the organisms have many months, years, or even decades to "get used" to the treatments. Seasonal or intergenerational responses could change the story quite a bit, particularly as different rates of acclimation and adaptation across different trophic levels alter the response of individual taxa.

Answer:

This is a very valid point. This experiment targeted the acute response rather than the long-term one. This sort of community experiments are, however, very difficult to keep stable for longer than a few weeks, and, as a result we currently have no real experimental design system as to how to deal with multi-annual responses to communities. We have included a comment in the text to deal with this important aspect (Ln 205).

In summary, I think this experiment presents some very interesting results that will be a valuable addition to the broader literature, but they do need to be carefully couched within the quite limited setting of the experimental conditions, duration, and trophic complexity that they were examined under.

Answer:

In the discussion, we toned down our words and are now more careful in our interpretation of the data (Ln 129, 164, 221).

Ln33. Change to either “greenhouse gas emissions” or “emissions of greenhouse gases”.

Answer:

Changed to “greenhouse gas emissions”.

Ln47. Change to either “copepod abundance” or “abundance of copepods”

Answer:

Changed to “copepod abundance”.

Ln52. “Amiable” is not the right word here...change to something like “conducive”

Answer:

Changed to “conducive”.

Ln100. There is statistical support for the first part of this sentence (related to *Guinardia flaccida*), but no such support for the second part about contribution of nanophytoplankton. Are these differences between treatments significant? It could also be made clearer whether the ambient was significantly different from both ERCP 6.0 and 8.5, and whether the latter two were or were not significantly different from one another.

Answer:

We included the statistical results for nanophytoplankton and showed the difference of the ERCP scenarios compared to the Ambient (Ln114).

Ln119. This sentence is too vague. “Affecting” primary production how and what are some of the potential “subsequent food-web effects”?

Answer:

We excluded this sentence for being excessively speculative.

Ln121. That’s a reasonable assumption, but it is just speculation since you cannot disentangle the independent effects of nutrients, warming, and acidification in this experiment. Try to be clear about that throughout and not over-interpret the role of individual stressors in your results. It would also help to provide stronger support from the literature for the role of P-limitation in lowering phytoplankton size and overall biomass.

Answer:

We changed the sentences to: “Since the phytoplankton bloom rapidly depleted DIP in the ERCP 6.0 and 8.5 scenarios (Supplementary Figure 2), we pose that the above-described phytoplankton biomass responses were mostly driven by DIP availability. The ERCP scenarios-induced smaller phytoplankton cell sizes are favourable for microzooplankton and as a consequence direct the flow of energy to the microbial food web, rather than efficiently fuelling higher trophic levels (Azam et al., 1983 and Legendre and Le Fèvre, 1995).”

Ln130. The discussion from here until Ln148 is interesting, although I am left wondering why the species may be viewed as a “canary” if there is ample research showing that it can actually thrive in acidified conditions. Could this be due to the combined effects of warming and nutrients on acidification, as you briefly allude to at Ln148? I think we need to see a little more development of that discussion, and particularly the evidence for why the coccolithophore species shows such variable responses to acidification.

Answer:

Due to the fact that calcification is a process highly sensitive to ocean acidification in many organisms, including coccolithophores, the high success in culturing *E. huxleyi* and the short period of time needed to acquire results from an experiment with it, this species has been extensively used to test ocean acidification effects. We included information on the effect of $p\text{CO}_2$ on calcification to be strain-specific in the case of *E. huxleyi*, but we have also included in the text now scientific evidence of increasing calcification under higher $p\text{CO}_2$ and temperature on this species. Although there is not much information on why different strains of *E. huxleyi* have different responses to ocean acidification, this response has been suggested to be modulated by other factors, such as temperature (Benner et al., 2013 and Sett et al., 2014).

Ln154. Again, why present just one overall statistic and not the pairwise comparisons that are actually presented in the preceding text? I'm thinking about the LRT equivalent of a Tukey post-hoc test here to see which treatments were significantly different from one another.

Answer:

Here we show the differences of ERCP scenarios compared to Ambient, as both were statically significant we included only one p -value. We changed the text to point out that both scenarios were significantly different from the Ambient.

Ln158. It's almost impossible to see the difference you describe here in Figure 3b. I suggest presenting separate barplots for each of the four taxa crammed into that figure, allowing you to put them each on sensible y-axis scales. It should be feasible to squeeze four small panels into the space currently occupied by Figure 3b, using a single y-axis label and colouring by treatment, rather than taxa for consistency with the other figures.

Answer:

We changed the figure as suggested.

Ln164. I find the discussion from here until Ln193 to be quite speculative and unclear. Metabolism was not measured, nor was nutritional demand quantified, so speculating on the role of these processes in the prevalence of a particular species (*Penilia avirostris*) feels quite tenuous. It also doesn't quite make sense to me, when the treatment where it was most abundant had an intermediate temperature and nutrient availability to the other two treatments. This indicates a hump-shaped response, and yet you said that temperatures were "well within the tolerance range" of the species. Have a rethink about the message in this section because it doesn't quite flow.

Answer:

Penilia avirostris was the second most abundant species of mesozooplankton during the experiment. The abundance of this cladoceran was between 30-120 individuals per Liter, while *N.*

scintillans was present up to 1,500 individuals per Liter. Therefore, we do not consider this species as important to the total mesozooplankton biomass as *N. scintillans*. It is known that *P. avirostris* and *N. scintillans* thrive in temperatures warmer than those we used in our experiment, thus we rather suggest that food availability was responsible for the hump-shaped response observed. We added this information to the text (Ln 184, 189).

Ln190. This is of course likely, but also quite speculative since you did not quantify the biomass of higher trophic levels in your experiment. A more complex trophic structure may have a whole host of interesting feedbacks (e.g. through altered top-down control) that could fundamentally change the conclusions of your experiment (which put a cap on the trophic pyramid at mesozooplankton, removing any role of top-down control on their biomass).

Answer:

As mentioned in our response above, we agree with the reviewer that including top-down control on mesozooplankton may yield different responses. We now explained these aspects and justify the exclusion of larger consumers, both in the material and methods as well as in the discussion (Ln 233, 383).

Ln196. Where is the statistical support for this statement? Note that it feels like a different statement to the subsequent two sentences, i.e. a comparison of treatments, versus statements about the trajectory of biomass through time within the various treatments. There is a general lack of transparency about the underlying statistics throughout the paper.

Answer:

We added the statistical results to this sentence showing a p-value of 0.055 for the effect of the ERCP scenarios on the diatoms biomass (Ln 229).

Ln218. I'm a little confused by the discussion here. If interaction strengths are expected to strengthen between microzooplankton and their bacterial prey, then why would bacterial biomass have increased in tandem with microzooplankton biomass in the ERCP 8.5 treatment, as indicated at Ln210? There are a lot of these little inconsistencies throughout the discussion that means the explanation of the results does not quite gel.

Answer:

The increase of bacterioplankton biomass in the ERCP 8.5 was likely linked to the decay of the phytoplankton bloom. Indeed, this has been shown to be associated with polysaccharide exudation which fuels bacterial growth, and, during these phases of growth, bacterial production and successions are often bottom-up controlled (<https://doi.org/10.7554/eLife.11888.001>). We clarified these aspects in the manuscript (Ln 284).

Ln220. I think "anodyne" may be the wrong choice of word here... "ecologically insignificant" perhaps?

Answer:

Changed to "ecologically insignificant".

Ln223. Why would intensified trophic pathways through microzooplankton diminish energy transfer efficiency? The logic is unclear and lacking support from the literature.

Answer:

We added the sentence to the discussion: “Strengthened energy flow through an additional trophic level leads to additional loss of organic carbon and, therefore, less efficient energy transfer to larger grazers (Fenchel, 2008 and Aberle et al., 2015).”

Ln274. It’s actually pretty surprising to see that the mesocosms were lined with plastic bags. Are there any concerns about this artificial substrate (flexible plastic seems even less realistic than the typical hard, smooth sides of a mesocosm), or indeed that it may act as a fourth stressor in the experiment (i.e. plastic pollution has been shown to significantly alter marine communities)?

Answer:

We do not expect the LDPE bags to be a fourth stressor to the planktonic organisms in our experiment. The material we used is even produced for the food industry for packaging and, although, zooplankton can actually ingest microplastic, only very high concentrations of microplastic have negative effects (<https://doi.org/10.1021/es400663f>).

Ln280. I think it should be made clear in the main text that the experiment actually just consists of three replicates of each of the three treatments...this really is quite a skeletal design and likely gives you very little statistical power to detect differences between treatments. That puts quite a bit of uncertainty over the story that is presented.

Answer:

We added this information in the main text of the manuscript. However, we wish to stress that, despite using three replicates, we identified striking responses and strong statistically significant differences between treatments, which reinforce the reliability of our results. We added this information to the text.

Ln284. It occurs to me that you are effectively taking ambient communities and then suddenly exposing them to water that is 1.5 or 3.0 °C warmer than they are used to. Could the “shock” of this sudden change lead to many of the effects that you subsequently see, and how much does this differ from a real climate change scenario, which will slowly ramp up over many years?

Answer:

The temperature in the mesocosm took about 24 hours to change and stabilize in the warmest scenario, this means 0.12°C increase per hour in the warmest scenario. We do not consider this warming rate to be a very rapid increase and it is well in the range of the temperature fluctuations which planktonic organisms may experience in coastal zones.

Ln321. The procedure does indeed seem to argue for a good likelihood of homogenous starting communities...but was this tested through sampling? It would be really great to show no treatment effects in the “before” communities to dispel any concerns here, and could convert this into a full before-after-control-impact style experimental design, which would be more powerful.

Answer:

We added results from the Kruskal-Wallis test showing no statistical difference in biomass of the four plankton groups across scenarios on the first day of the experiment (Ln 402).

Ln359. I don’t want to make too big a deal of this, but it is a little concerning that the acidification treatment did not actually match the IPCC scenarios. Similarly, it seems like the nutrient concentrations were not all that different (even at the beginning of the experiment for the two ERCP scenarios). It might be worth acknowledging that in the main text, i.e. the goal was to imitate IPCC scenarios, but this was not perfectly realised in the experiment.

Answer:

The reviewer is correct, we acknowledge on line 368 that the pCO₂ levels were lower than the IPCC scenarios. Phosphorus concentrations were well within the experimental design, as they follow the ratio to the natural low nitrogen concentration during the period of the experiment.

Ln431. The description of the statistical analysis here seems to indicate that the LRT only distinguishes a significant difference between the ambient control and the combined ERCP 6.0 and 8.5 scenarios. Is that the case, or have I misinterpreted the description? If the latter, it needs a better description...if the former, it seems like this is not a very good test for determining treatment differences, since we cannot tell the difference between the two ERCP scenarios. I did not get that impression from the main text, although I was frustrated by the lack of statistical clarity on which treatments were different from each other, so something needs to be improved here (either in the statistics or in the presentation and description).

Answer:

Yes, we are not statistically comparing the two ERCP scenarios. Since our experimental design targets the influence of future scenarios, our approach here is to see if there is an effect of the ERCP scenario compared to the Ambient scenario. We agree that our precedent way to present statistics about GLM were confusing. In a nutshell, we tested the effect of treatment by a GLM. The second step was to compare a model with only scenario, with a model with scenario + time to see if time was statistically improving the model, meaning differences in timing between treatments. It permitted us to ensure either that bloom dynamic was similar or not among treatments, or, that the changes seen over the experiment period are caused by the scenario and not by time. We added this information to the text. We improved the paragraph to present more information about the statistical analysis.

Reviewer #3 (Remarks to the Author):

Dear authors,

I have read the following manuscript „Structural alterations of planktonic food webs under global change: an integrated multiple driver mesocosm experiment” carefully and find it very well done and represents a good contribution to this field of knowledge. I liked the overall manuscript very much and my comments tend not to be detailed but rather a bit broader. While the data is nicely analysed and presented with the appropriate statistical tests, I only have some concerns about the methodology used. However, I think this can be discussed very easily by you. The water used in the mesocosms for the experiment was taken directly from the sea. My first question is from what depth was this taken? Was this a mixed sample from several depths or only water from the upper layers. This would be very interesting, especially as the depths of the mesocosms are quite limited. This brings me to the next comment. The mesocosms described here are shallow, and given the very great depth of the upper layer in the free ocean, this could have an influence on the course of the experiment and thus on the data collected. Specifically, I mean that a very complex community and its interrelationships were observed in a rather unusual part scenario. It would be very good if you could briefly address this issue. Because this setup represents another stress factor, that of lower mixing depth. Mesocosm experiments with shallow experimental depth can lead to particular turn over events. This should definitely be discussed in this manuscript, especially as you relate the results onto future climate change scenarios. If this experimental problem were discussed, it would support the weighting of the results even more.

Answer:

We thank the reviewer for the positive assessment of our work. The seawater for this experiment was collected from the upper 5 meters layer, and the temperature in the field was measured daily at the same depth. We have included the information of depth in the manuscript where we describe water collection procedures (Ln 391).

As you describe, there was a temperature control and temperature was regularly compared with the open sea and accordingly adjusted in approaches with increased temperature. However, I wonder if there was not also a temperature increase in the ambient treatments, as the mesocosm facility is land-based, and if so, how was this compensated for or included in the data analysis?

Furthermore, I think the manuscript is very well written and the data is certainly of broad interest to a wide readership.

Answer:

The mesocosm system was designed to keep temperature stable through cooling/heating devices following the temperature measured in the sea. The system worked perfectly and no deviation of temperature was seen over the course of the experiment. We also present this information on Supplementary Material 6c.

Please let us know if you would like to further discuss this matter and if you have any more questions. We will be glad to answer them.

Yours sincerely,

Hugo Duarte Moreno

On behalf of the co-authors (Martin Köring, Julien Di Pane, Nelly Tremblay, Karen H. Wiltshire, Maarten Boersma and Cédric L. Meunier).

REVIEWERS' COMMENTS:

Reviewer #1 (Remarks to the Author):

The authors addressed my previous concerns. The manuscript has improved.

Please consider discussing top-down vs bottom-up control, regarding the increase of *E.hux*. Maybe the decrease of large grazers could explain it, rather than resources:

Behrenfeld, M.J., Boss, E.S. & Halsey, K.H. Phytoplankton community structuring and succession in a competition-neutral resource landscape. *ISME COMMUN.* 1, 12 (2021).

<https://doi.org/10.1038/s43705-021-00011-5>.

Reviewer #2 (Remarks to the Author):

I would like to commend the authors on a generally sound revision of their manuscript. The additional sentences acknowledging some of the limitations of the study will be helpful to avoid some readers over-interpreting the results and I appreciate the clarification on some of the statistical fine points. I think the authors have also provided a comprehensive response to the main criticism of R1, adding information on the picophytoplankton, nanociliates, and nanoflagellates to the methodological text and including relevant new literature in their paper. They have also addressed the methodological queries of R3.

I would be happy to see this paper published in *Communications Biology* following some minor revisions related to my outstanding comments below. As a general note, it would be helpful to include direct quotations to revised text throughout the response document in future revisions (these were omitted in the first few pages). I found the cross-referencing tiresome and the line numbers sometimes did not seem to match up.

In the first response to R2, the authors note the rarity and importance of multi-driver (a.k.a. multiple stressor) experiments. I couldn't agree more! I also appreciate that single stressor experiments are not that useful on their own, but the gold standard really is the combination of all single and multiple stressors. Here, we can see the impact of all the stressors together, but also tease apart the individual contribution of each stressor to the overall effect from the remaining treatments. This should be acknowledged in the text.

The separation of taxa into different panels in the new Figure 3b is great, although this would probably now be best labelled as panels b, c, d, and e.

The response mentioning the addition of statistical support and a p-value of 0.055 at Ln229 does not seem to have been made in the manuscript. This should be rectified. First of all, note that the line number should be Ln207 (one of several examples of the line numbers not matching up between response document and revised text). The p-value is also greater than 0.05, so the authors should acknowledge this with a statement such as "a marginally non-significant increase in" rather than "a gradual increase in".

For the comment regarding Ln274 and the plastic used in the experiment, it would be helpful to also add a note on this in the text to allay any potential concerns from readers (as I had) and perhaps also to offer a reference to other mesocosm studies that employ this type of design.

For the comment on Ln321, it would also be helpful to add this information to the main text. Basically, please try to change something in the text rather than just answering directly to the reviewers. We raise concerns that any reader could also share, so it is important to add your very relevant points about why we shouldn't be concerned to your manuscript.

Reviewer #3 (Remarks to the Author):

Dear author,

I read your revised manuscript and Structural alterations of planktonic food webs under global change: an integrated multiple driver mesocosm experiment. In general, you addressed all comments of all reviewer well. However, I am somewhat skeptical about the identification of nanoflagellates using a light microscope.

Communications Biology
Revision

Hugo Duarte Moreno
Alfred Wegener Institute
Helmholtz Centre for Polar and Marine
Research
Ostkaje 1118, 27498
Helgoland, Germany
hugo.moreno@awi.de
tel: +49 172 2620219

December 21st 2021

Dear Reviewers,

Thank you for your interest in our work. We appreciate the positive feedback and acknowledge the relevant insights in improving our manuscript.

Please find below our remarks and answers to your questions.

Reviewer #1 (Remarks to the Author):

The authors addressed my previous concerns. The manuscript has improved. Please consider discussing top-down vs bottom-up control, regarding the increase of *E.hux*. Maybe the decrease of large grazers could explain it, rather than resources: Behrenfeld, M.J., Boss, E.S. & Halsey, K.H. Phytoplankton community structuring and succession in a competition-neutral resource landscape. *ISME COMMUN.* 1, 12 (2021). <https://doi.org/10.1038/s43705-021-00011-5>.

Answer:

*We have included on Ln205 the following information acknowledging the possible top-down control effect on *Emiliana huxleyi*: "In parallel to bottom-up effects and to a lesser extent, we expect that the lower grazing pressure from mesozooplankton might also have contributed to the increase of *Emiliana huxleyi* in the ERCP 8.5 scenario (Behrenfeld et al., 2021)."*

Reviewer #2 (Remarks to the Author):

I would like to commend the authors on a generally sound revision of their manuscript. The additional sentences acknowledging some of the limitations of the study will be helpful to avoid some readers over-interpreting the results and I appreciate the clarification on some of the statistical fine points. I think the authors have also provided a comprehensive response to the main criticism of R1, adding information on the picophytoplankton, nanociliates, and nanoflagellates to the methodological text and including relevant new literature in their paper. They have also addressed the methodological queries of R3.

I would be happy to see this paper published in *Communications Biology* following some minor revisions related to my outstanding comments below. As a general note, it would be helpful to include direct quotations to revised text throughout the response document in future revisions (these were omitted in the first few pages). I found the cross-referencing tiresome and the line numbers sometimes did not seem to match up.

In the first response to R2, the authors note the rarity and importance of multi-driver (a.k.a. multiple stressor) experiments. I couldn't agree more! I also appreciate that single stressor experiments are not that useful on their own, but the gold standard really is the combination of all single and multiple stressors. Here, we can see the impact of all the stressors together, but also tease apart the individual contribution of each stressor to the overall effect from the remaining treatments. This should be acknowledged in the text.

Answer:

We have included on Ln71 the following information acknowledging the focus of our experimental design: "It is currently of utmost importance to make accurate and reliable predictions of the fate of planktonic communities in future conditions. Although our experimental design does not enable to draw conclusions about individual drivers effect, we believe that our work provides a more realistic assessment of these drivers' impact than an experiment addressing drivers singly would."

The separation of taxa into different panels in the new Figure 3b is great, although this would probably now be best labelled as panels b, c, d, and e.

Answer:

We have changed the figure accordingly.

The response mentioning the addition of statistical support and a p-value of 0.055 at Ln229 does not seem to have been made in the manuscript. This should be rectified. First of all, note that the line number should be Ln207 (one of several examples of the line numbers not matching up between response document and revised text). The p-value is also greater than 0.05, so the authors should acknowledge this with a statement such as "a marginally non-significant increase in" rather than "a gradual increase in".

Answer:

*We believe there is some misunderstanding about this comment due to the different versions of the manuscript (marked and unmarked). The statistical information mentioned above is described in the text on Ln200 (current marked version), regarding the difference in the proportion of diatom biomass across scenarios. We have also changed the text to comply with the comments above: "Hence, the collapse of *N. scintillans* may be driven by a marginally non-significant increase from Ambient, to ERCP 6.0, to ERCP 8.5, in the proportion of diatoms within the phytoplankton community (LRT, df 86, $p = 0.05$)". The statistical information regarding differences of microzooplankton biomass across scenarios is displayed on Ln216 (current marked version).*

For the comment regarding Ln274 and the plastic used in the experiment, it would be helpful to also add a note on this in the text to allay any potential concerns from readers (as I had) and perhaps also to offer a reference to other mesocosm studies that employ this type of design.

Answer:

We have added on Ln316 the following information: "The LDPE was chosen as material for the bags as it should not represent a risk for planktonic organisms either, since it also is used in food industry for packaging."

For the comment on Ln321, it would also be helpful to add this information to the main text. Basically, please try to change something in the text rather than just answering directly to the reviewers. We raise concerns that any reader could also share, so it is important to add your very relevant points about why we shouldn't be concerned to your manuscript.

Answer:

We have added on Ln87 the information present in the materials and methods to highlight the homogeneity of scenarios on the first day of the experiment: "Across scenarios, no significant difference was found in biomass of phytoplankton, microzooplankton and bacterioplankton on the first day of the experiment (Kruskal-Wallis test, $df = 2$, $p > 0.05$)."

Reviewer #3 (Remarks to the Author):

Dear author,

I read your revised manuscript and Structural alterations of planktonic food webs under global change: an integrated multiple driver mesocosm experiment. In general, you addressed all comments of all reviewer well. However, I am somewhat skeptical about the identification of nanoflagellates using a light microscope.

Answer:

We understand your concern about the identification of nanoflagellates through light microscopy, but we assure that we only quantified and grouped them by size class. We did not identify nanoflagellates to species level due to the limitation of this method on this group.

Please let us know if you would like to further discuss this matter and if you have any more questions. We will be glad to answer them.

Yours sincerely,

Hugo Duarte Moreno

On behalf of the co-authors (Martin Köring, Julien Di Pane, Nelly Tremblay, Karen H. Wiltshire, Maarten Boersma and Cédric L. Meunier).